# An Efficient Automatic Meta-Path Selection for Social Event Detection via Hyperbolic Space

## ABSTRACT

Social events reflect changes in communities, such as natural disasters and emergencies. Timely detection of these situations can help residents and organizations in the community avoid danger and reduce losses. The complex nature of social messages makes social event detection on social media challenging. Existing methods usually construct social messages into heterogeneous information graphs to facilitate learning their semantic and structural information. However, they usually assume a fixed set of meta-paths, which often cannot describe real-world data sets well. On the other hand, a large number of social messages are not labeled due to expensive labeling work, which leads to an increase in model training costs. In order to solve the above challenges, we proposed a Heterogeneous Information Graph representation via Hyperbolic space combined with an Automatic Meta-path selection (GraphHAM) model, an efficient model that automatically selects meta-path and combines hyperbolic space to learn information on social media. In particular, we apply an efficient automatic meta-path selection technique and convert the selected meta-path into a vector. We then designed a novel Hyperbolic Multi-Layer Perceptron to further learn the semantic and structural information of social information. Extensive experiments show that GraphHAM can achieve outstanding performance on real-world data using only 20% of the whole dataset as the training set.

## CCS CONCEPTS

• **Computing methodologies** → **Artificial intelligence**; • **Information systems** → **Social networks**.

## KEYWORDS

Social Event Detection, Graph Neural Networks, Automatic Meta-Path, Hyperbolic Space

## 1 INTRODUCTION

Social events can be defined as hot topics that people discuss on social media [6]. It reflects a major impact on communities, such as natural disasters and emergencies, and also reflects people's attitudes and reactions to some events, such as presidential elections and institutional reforms. Social event detection extracts these influential events from the vast reservoir of online information, making it a vital and indispensable process in understanding contemporary societal dynamics. Today's ubiquity of mobile devices means social media platforms such as Twitter, Facebook, and Weibo are often the first to witness events as they occur. Therefore, social media often serves as an important source of information for relief organizations and governments on emergencies and natural disasters, allowing them to detect disasters at an early stage, monitor the development of disasters, and carry out subsequent rescue operations [21]. Based on the above reasons, social event detection in social media has gained significant attention in both academic and industrial circles [2, 6, 9, 22, 27, 40].

The difficulty of social event detection lies in text classification or clustering, specifically classifying or clustering similar messages into the same topic [2, 16]. However, unlike traditional text classification, classifying text from social media has the following three challenges: (1) Social media messages are not just text, usually containing some other heterogeneous information, such as time, places, people, and entities. In order to preserve the characteristics of social media messages as much as possible, a common practice involves building social media messages into Heterogeneous Information Networks (HINs) [29]. Some methods[6, 22, 23] convert the HINs into homogeneous graphs of specific patterns that are still relatively rouge. The semantic features and structural features rich in heterogeneous information graphs are complex, which makes it difficult for the existing models to extract useful features from HINs. The more advanced way to deal with HINs is to use meta-path [31], which can intuitively exploit structural information in heterogeneous graphs. However, most of the existing models [35, 39] use fixed meta-path sets, hindering their capacity to adequately express real-world datasets. Thus, how to learn features from HINs is still a thorny problem. (2) The spread of social media is mainly based on mentions and forwardings between people, resulting in social media data having a tree structure [1]. The data in the tree structure grows exponentially. However, existing social event detection models are generally designed based on Euclidean space, and they cannot capture tree structure data well [13, 37]. Therefore, how to better learn features from tree-structured data is also a problem to be solved. (3) Social media generates volumes of unlabeled data, but the high costs associated with manual labeling make it impractical to annotate social media data in large quantities. Therefore, a challenge lies in effectively training models with limited labeled datasets.

In this paper, we propose a Heterogeneous Information Graph representation via Hyperbolic space combined with an Automatic Meta-path selection (GraphHAM) model, which combines a graph neural network with efficient automatic meta-path selection technology and hyperbolic space representation to solve the above challenges. In particular, we only enumerate all meta-paths from the selected points instead of enumerating meta-paths from all nodes in the dataset, which greatly improves our efficiency. After that, we embed the meta-path of each category through a hyperbolic space representation model to solve the problem of heterogeneity. Next, an attention mechanism is utilized to select the meta-path. Finally, since we filtered out some nodes without significant connections when constructing the heterogeneous information graph, we embedded all text features through hyperbolic space as feature supplements for the filtered-out nodes. The contributions of our work can be summarized as follows:

- We propose GraphHAM, an efficient heterogeneous information graph learning framework combining the automatic selection of meta-paths from the selected points and hyperbolic spaces representation for learning tree-structured data.
- We solve the problem that the existing HIN-based social media detection model manually sets the meta-path by efficient node sampling and automatic meta-path selection. We also apply hyperbolic space representation models in this study to reduce the distortion of embedded tree-structured data, which is ignored by existing social event detection models. Verify which hyperbolic space model is suitable for the dataset from social media.
- Extensive experiments show that GraphHAM is competitive and improved compared to baseline models, achieving good performance using only 20% of the data volume as the training set.

## 2 RELATED WORK

### 2.1 Social Event Detection

The development of social event detection models starts from social event detection models based on topic detection, such as LDA [5] and TF-IDF [28]. This type of model mainly focuses on classifying similar texts into the same topic and calculates the similarity between them by counting the co-occurring words that appear in the text. However, this method is largely limited by the sparsity of keywords, especially short texts on social media such as Facebook and Twitter. The large number of short texts appearing on social media has greatly reduced the performance of this type of model in social media. To solve this problem, the researchers try to let the model understand the meaning behind the words and let the model learn the semantic information of the words. The Word2Vec [19] is their method to let the model understand the semantics of the words. Based on Word2Vec, researchers proposed GPU-DMM [14] further to improve the model's performance in short text classification. However, the semantic improvement behind simply understanding words is limited, and the sparseness of words is still a key issue that hinders model improvement. It still needs some additional information to supplement the information loss caused by word sparseness. Therefore, researchers proposed SGNS [30], which takes the relationship between words into model learning. This model solves the problem of sparse words in short texts by learning the semantics of the entire sentence.

However, traditional social event detection models based on topic detection only focus on text, and they ignore the essence of social media: heterogeneity. If the model only learns text information, it will lose a lot of structural information and semantic information between entities. Therefore, researchers try to construct social media data into a heterogeneous information network to retain as much useful information as possible. Based on our knowledge, PP-GCN [22] applies HIN to social media detection for the first time. It learns the semantic information and structural information on HIN through meta-path [31]. However, the meta-path set on PP-GCN is designed manually, which has limitations and uncertainties. Even if the meta-paths are designed by experienced experts, they cannot perfectly describe real-world data. KPGNN [6] and FinEvent [23]

are the latest HIN-based social event detection models proposed two years later, but they focus on incremental social event detection. Their methods for learning HIN are still similar to PP-GCN and have not been improved. Therefore, learning more correct features from HIN is still a thorny problem that needs to be solved.

### 2.2 Automatic Meta-path Selection

The problem with meta-path is that it requires manual design by experts based on prior knowledge. This design cannot perfectly describe real-world data. In addition, the importance of each meta-path in the data set is also different, and it is difficult to manually define the weight of each meta-path. Therefore, how to automatically select a meta-path from HIN has become a research hot spot in many fields. Wei [36] proposed a model in 2018 that selects the importance of meta-paths by maintaining the proximity between nodes. In the same year, Wang [33] proposed an unsupervised meta-path selection model that uses a minimum spanning tree to rank meta-path importance. However, both of these problems only consider the problem of information retrieval and do not consider representation learning. GTN [38] is a GCN representation learning model combined with meta-path selection. GTN selects the meta-path based on the graph transformer layer. However, the meta-path selection of the GTN model is performed on all nodes, which means that all nodes need to be enumerated, which is very resource-consuming. In addition, all nodes are projected in the same feature space, which cannot distinguish the features of different nodes well.

### 2.3 Hyperbolic Space Representation

Another disadvantage of the meta-path is that its length is limited. For data with a tree structure like social media data [25], it is difficult to capture long-distance root-to-leaf relationships. On the other hand, although embedding learning in Euclidean space has achieved great success, embedding tree-structured data into Euclidean space will cause distortion [12]. Since the data growth of tree structure is exponential, but Euclidean space is not, this will cause the leaves far away from the root to be very close and unrecognizable. Unlike Euclidean spaces, the growth of hyperbolic spaces is also exponential. Therefore, hyperbolic space is ideal for embedding tree-structured data [10, 12]. However, there are no basic statistical algorithms in hyperbolic spaces, such as calculations of vectors and matrices [12]. Therefore, some basic algorithms cannot be applied in hyperbolic spaces. HNN [12] applies simple neural networks such as RNN to hyperbolic space, but it only focuses on the structural information of the graph and ignores node features. HGCN [7] applies GCN to hyperbolic space based on HNN and implements GCN's aggregation of nodes on hyperbolic space. Nevertheless, existing hyperbolic space learning is based on homogeneous information networks, and there are still some shortcomings in the application of heterogeneous information networks. Therefore, we need to design a framework to apply hyperbolic space in the heterogeneous social event detection environment.

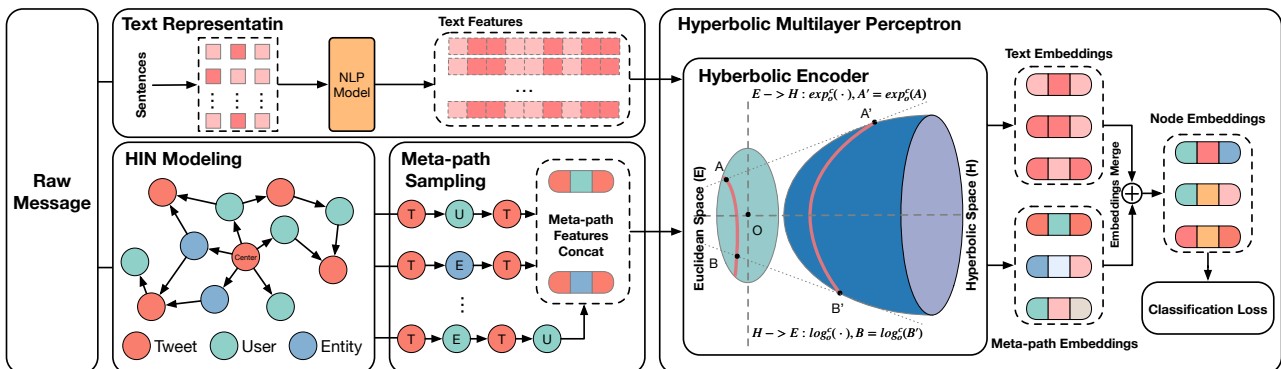

**Figure 1: An overview of GraphHAM. GraphHAM contains four main parts: (a) HIN Modeling (b) Meta-path Sampling, (c) Nodes Embedding, and (d) Selection via Attention.**

## 3 PRELIMINARIES

In this section, we summarize the concepts related to the background of our work, including HIN, Meta-path, and hyperbolic space representation.

**Definition 1** HIN. A HIN is defined a graph $\mathcal{G} = (V, E, \mathcal{N}, \mathcal{E}, \mathcal{X})$. Here $V$ and $E$ are the set of nodes and edges. $\mathcal{N}$ and $\mathcal{E}$ denote the sets of node and edge types where $|\mathcal{N}| > 1$ and $|\mathcal{E}| > 1$. $\mathcal{X}$ is the set of node features. Furthermore, there are two functions $\Phi : V \rightarrow \mathcal{N}$ and $\Psi : E \rightarrow \mathcal{E}$ mapping nodes and edges to their types.

**Definition 2** Meta-path and Meta-path Instance. A meta-path $\mathcal{P} = n_1 \xrightarrow{e_1} n_2 \xrightarrow{e_2} \cdots \xrightarrow{e_l} n_{l+1}$, where $n_i \in \mathcal{N}$ and $e_j \in \mathcal{E}$ and $l$ is the length of this meta-path. It describes the relation $e = (e_1, e_2, \cdots, e_l)$ between node types $n = (n_1, n_2, \cdots, n_{l+1})$. A meta-path instance $p$ is a real path that follows a certain meta-path $\mathcal{P}$ in the heterogeneous graph $\mathcal{G}$.

**Definition 3** Hyperbolic Representation. The aim of hyperbolic representation is to map the features from Euclidean space to hyperbolic space. We have two types of hyperbolic embedding models: $\mathbb{P}$ for the PoincareBall model and $\mathbb{H}$ for the Hyperboloid model. For the PincareBall model, we denote $E_o \mathbb{P}^{d,c}$ as the Euclidean space and $\mathbb{P}^{d,c}$ as the hyperbolic representation via the PoincareBall model, where $o$ is the center of the space, $d$ is the dimensions, and $c$ is the curvature of this space. Thus, the mapping process from Euclidean space to hyperbolic space is $exp_o^c : E_o \mathbb{P}^{d,c} \rightarrow \mathbb{P}^{d,c}$, and the opposite mapping is $log_o^c : \mathbb{P}^{d,c} \rightarrow E_o \mathbb{P}^{d,c}$. Specifically, for a node $a \in E_o \mathbb{P}^{d,c}$ and $a' \in \mathbb{P}^{d,c}$, we have: $exp_o^c(a) = a'$ and $log_o^c(a') = a$, where

$$exp_o^c(a) = tanh(\sqrt{c}\|a\|) \frac{a}{\sqrt{c}\|a\|}, \qquad (1)$$

$$log_o^c(a') = artanh(\sqrt{c}\|a'\|) \frac{a'}{\sqrt{c}\|a'\|}. \qquad (2)$$

For the hyperboloid model, we denote $E_o \mathbb{H}^{d,c}$ as the Euclidean space and $\mathbb{H}^{d,c}$ as the hyperbolic representation via the hyperboloid model and the $exp_o^c$ and $log_o^c$ functions are defined as:

$$exp_o^c(x) = cosh\left(\frac{\|x\|}{\sqrt{c}}\right) y' + \sqrt{c} \cdot sinh\left(\frac{\|x\|}{\sqrt{c}}\right) \frac{x}{\|x\|}, \qquad (3)$$

$$log_o^c(x') = d_{\mathbb{H}}^c(x', y) \frac{y' + \frac{1}{c}\langle x', y'\rangle_{\mathcal{M}} x'}{\|y' + \frac{1}{c}\langle x', y'\rangle_{\mathcal{M}} x'\|}, \qquad (4)$$

where $x', y' \in \mathbb{H}^{d,c}$, $x \in E_o \mathbb{H}^{d,c}$ with $x' \neq y', x \neq 0$, and $d_{\mathbb{H}}^c(\cdot)$ is the function calculates the distance between two nodes in hyperbolic space and $\langle ., .\rangle_{\mathcal{M}}$ is the Minkowski inner product.

## 4 METHODOLOGY

In this section, we introduce our frame Heterogeneous Information Graph representation via Hyperbolic space combined with an Automatic Meta-path selection, called GraphHAM, an overview of this model shown in Fig. 1. In general, our framework contains two main models: The text Model for text feature extraction and the Heterogeneous Information Network(HIN) Model for HIN structural information learning. Specifically, Our model mainly follows four main steps:

- First, we process the original data, count different points and edges, assign them features, and finally build a HIN based on the relationship between points and edges.
- Then, we randomly select a part of the nodes $(V_s)$ in the HIN, and then build a meta-path set from these sample nodes.
- We design the Hyperbolic Multilayer Perceptron (HMLP) corresponding to different types of meta-paths, project them into equal-length vectors and use an attention layer to calculate the weight of each meta-path and the embedding of nodes.
- In addition, we use additional HMLP to learn all text information as a supplement to the information deleted in constructing HIN. Finally, we combine the nodes embedding together and calculate the loss.

### 4.1 HIN Modeling

For data preprocessing, our main purposes are (1) to make full use of social media data by extracting different types of elements from messages, and (2) to construct relationships between different types of elements. In order to meet the above purposes, we adopt a HIN, which is a graph that contains multiple nodes and the relationships between nodes. It can express different types of data in social media and the relationships between them. Let's take Twitter data as an example. When given a tweet $t_i$, we will extract the text information in $t_i$, delete special characters, emoji, and URL, and extract the named entities $e_j$ and the rest of the words after removing the stop

words in the text. We will put $t_i$ and $e_j$ into HIN as a type of nodes, and add a type of edge between them. In order to better describe social media, we add users $u_k$ as a node to HIN and connect them with the tweets they have sent or the users mentioned by tweets.

The current three types of nodes do not have feature vectors yet, so the next step is assigning them features. For the features of tweets, we use the pre-trained 300-d GloVe [26] vectors [1] to convert the words extracted from the tweets into feature vectors and assign them to the tweet node. In addition, as time is an indispensable feature, we also convert it into a feature vector and combine it with the word feature vector to assign it to the tweet node.

For entity nodes, we use similar operations as tweet nodes to convert entities into feature vectors using the pre-trained 300-d GloVe. The difference is the way it combines time vectors. Since there will be a large number of repeated entities in different tweets, and they are distributed at different time nodes, we convert all the time nodes where this entity appears into feature vectors and then add them to take the mean. This aims to better estimate the time when people are discussing the event a lot.

For user features, based on the protection of user information, we did not extract any information about the user, including but not limited to the user's name, gender, address, beliefs, followers, and other information related to the user. All we have is the user's ID. Therefore, we connected the tweets associated with each user ID, filtered out users who were only connected to one tweet, and the features of the remaining users will be the average of the features of the tweets associated with them. In this way, we can also extract what events the user is more concerned about when there is little user information. After we operate on all tweets, HIN for social media data is constructed.

## 4.2 Meta-path Sampling

As mentioned before, GTN's automatic meta-path is enumerated from all nodes. Inspired by GraphMSE [17], we adopt the meta-path sampling part of GraphMSE. We only collect meta-path instances from the sampled node set $V_s$, where $V_s \subset V$. Then, the collection of meta-path instances is not an enumeration. We intentionally discard some meta-path instances by limiting the number of neighbor nodes searched, shown in Fig.1. This alleviates over-fitting and over-smoothing caused by exploring all neighbor nodes [15]. Experiments show that the model can be trained well when the sampling node set reaches 20% of all nodes.

## 4.3 Nodes Embedding

This section introduces how we embed meta-path instances into feature vectors. First, we introduce the HMLP, which contains the hyperbolic encoder inspired by HGCN [7]. We then introduce how to embed node features from text models and HIN models into our framework using HMLP in Section $4.3.2 - 4.3.4$.

*4.3.1 Hyperbolic Multilayer Perceptron (HMLP).* In order to better learn tree-structured data, we use hyperbolic space as the low-dimensional space for embedding. However, as mentioned before, there is no vector calculation in hyperbolic space, so we cannot

directly apply MLP in hyperbolic space. Therefore, for the implementation of HMLP, we need to project the hyperbolic embedding into Euclidean space for vector operations. For single-layer perceptron in Euclidean space, we have

$$\mathbb{E} = \sigma(\mathcal{W}X + b), \tag{5}$$

where $\mathcal{W}$ is a $V \times V$ weight matrix, $X$ is the node feature on Euclidean space, $\sigma$ is the activate function and $b$ is the bias. Based on the mapping relationship shown in the **Definition 3**, the vector operations on hyperbolic space like:

$$\mathcal{W} \otimes \mathcal{X} = exp_o^c(\mathcal{W}log_o^c(\mathcal{X})), \tag{6}$$

and

$$\mathcal{X} \oplus b = exp_o^c(log_o^c(\mathcal{X}) + b)), \tag{7}$$

where $\mathcal{X}$ is the node feature on hyperbolic space, $o$ is the center of the space, and $c$ is the curvature of the space. Thus, the single-layer perceptron on hyperbolic space is:

$$\mathbb{H}_{SLP} = exp_o^c(\sigma_1(log_o^c(\mathcal{W}_1 \otimes \mathcal{X} \oplus b_1))), \tag{8}$$

after that, we can deduce that the output of the two-layer perceptron is:

$$\mathbb{H}_{2LP} = exp_o^c(\sigma_2(log_o^c(\mathcal{W}_2 \otimes \mathbb{H}_{SLP} \oplus b_2))). \tag{9}$$

*4.3.2 Meta-path Embedding.* For a type of meta-path $\mathcal{P}_i$, there are several meta-path instances $p_j \in \mathcal{P}_i$ When we explore the neighboring nodes of point $v$. However, the features between different types of nodes are heterogeneous, so we cannot simply sum or average these features. Therefore, we concatenate these features according to order [11]:

$$X_p = CONCAT(X_{p_1}, X_{p_2}, \cdots, X_{p_n}). \tag{10}$$

Then, since there are meta-paths of different lengths and types, we use different HMLPs to correspond one-to-one with each type of meta-path to align the output vectors. For a type of meta-path $P_i$ from node $v$, we have:

$$\mathbb{H}_{\mathcal{P}_i}(v) = \sum HMLP_{\mathcal{P}_i}(CONCAT(X_{p_1}, X_{p_2}, \cdots, X_{p_n})), \tag{11}$$

where, $\mathcal{P}_i = (p_1, p_2, \cdots, p_n)$. Then, we need to map this hyperbolic representation to Euclidean space for the downstream task:

$$H_{\mathcal{P}_i}(v) = \mathcal{D}(\mathbb{H}_{\mathcal{P}_i}(v)), \tag{12}$$

where $\mathcal{D}(\cdot)$ is a decoder for hyperbolic embedding.

*4.3.3 Attention Layer.* We adapted graph attention from GAT [32]. For a node $v$, we have:

$$h_v = \mathcal{W}_1 X_v + \sum_{\mathcal{P}_i \in \mathcal{P}} \delta_i H_{\mathcal{P}_i}(v), \tag{13}$$

$$\delta_i = \frac{1}{|V|} \sum_{v \in V} \delta_{vi}, \tag{14}$$

$$\delta_{vi} = \frac{exp(-e_{vi})}{\sum_i exp(-e_{vi})}, \tag{15}$$

$$e_{vi} = LeakyReLU(\delta^T tanh([h_v||H_{\mathcal{P}_i}(v)])), \mathcal{P}_i \in \mathcal{P}. \tag{16}$$

Now, we have the HIN node representation $h_v, v \in V_s$ from $\mathcal{G}$ with attention weights of meta-paths $\alpha_i, \mathcal{P}_i \in \mathcal{P}$. The whole HIN representation is:

$$H_h = \sum_{v \in V_s} h_v \tag{17}$$

*4.3.4 Nodes Embedding Merge.* Since we filter out users who are only related to one tweet during HIN modeling, we will delete some nodes, resulting in the loss of some information. So in this part, we use the text information of the tweet as the input of HMLP and learn the representation of all tweet nodes:

$$\mathbb{H}_t = HMLP(X_t), \tag{18}$$

$$H_t = \mathcal{D}(\mathbb{H}_t), \tag{19}$$

where $H_t$ is the set of tweets representation, and $X_t$ is the set of tweets features. Finally, we generate the final nodes embedding by combining the $H_h$ and $H_t$:

$$H_f = H_h \oplus H_t, \tag{20}$$

here, $\oplus$ is the method for combining these embeddings, it could be sum, mean, or contact. The performance of these methods will be demonstrated in the ablation studies.

## 4.4 Objective Function

The objective function we use in this work is cross-entropy with labels $L$:

$$L' = softmax(H_f), \tag{21}$$

$$\mathcal{L}_{GraphHAM} = -\sum_{i=0}^{n} l_i log l'_i, \tag{22}$$

where, $L'$ is the prediction labels from final nodes embedding after sofmax, and $l_i \in L, l'_i \in L'$.

## 5 EXPERIMENTS

In this section, we will present and discuss the experimental results we designed. Our experiments focus on the impact of point classification on our model under different settings. The experiments are designed to answer several research questions:

- RQ 1: Is our model competitive compared to the baseline models?
- RQ 2: How does the training ratio impact our model?
- RQ 3: Can hyperbolic space improve model performance?
- RQ 4: What role do text models and HIN models play in the framework?
- RQ 5: How to choose the node embedding combination method?
- RQ 6: What impact do hyperparameters have on the model?
- RQ 7: Is our model more efficient than the baseline model?

## 5.1 Experimental Settings

*5.1.1 Datasets.* The main target domain of our model is social media, so we collected three real datasets through the Twitter API: the Twitter2012 [18], CrisisLexT6 [20], and Kawarith [3] datasets. The statistics of these datasets are shown in Table 1.

**Table 1: The statistical information of datasets.**

| Dataset | Nodes | Features | classes |
|---|---|---|---|
| Kawarith | 4,860 | 302 | 6 |
| CrisisLexT6 | 18,157 | 302 | 6 |
| Twitter2012 | 68,841 | 302 | 503 |

We list the node type of these datasets, and after HIN modeling, the statistics of HINs of these three datasets are shown in Table 2:

- **Kawarith**: Tweet(T), Entity(E), User(U).
- **CrisisLexT6**: Tweet(T), Entity(E), User(U).
- **Twitter2012**: Tweet(T), Entity(E), User(U).

**Table 2: The statistical information of HINs.**

| Dataset | Nodes | Edges | Meta-path |
|---|---|---|---|
| Kawarith-HIN | 9,743 | 14,181 | TE, TU, TUT, TET |
| CrisisLexT6-HIN | 31.625 | 41,415 | TE, TU, TUT, TET |
| Twitter2012-HIN | 98,070 | 148,979 | TE, TU, TUT, TET |

*5.1.2 Baseline.* We compare our model with the following baselines:

- **TF-IDF** (2003) [28]: is based on the importance of word frequency and is widely used in data mining and text data recommendation [4];
- **Word2Vec** (2013) [19]: is a method used by many social media detection models to understand the meaning behind the text;
- **FastText** (2016): is a word vector calculation and text classification tool open-sourced by Facebook in 2016. It is not very innovative in academic terms, but its advantages are also very obvious. In text classification tasks, FastText can often achieve accuracy comparable to that of deep networks, but its training time is many orders of magnitude faster than deep networks.
- **Bert** (2018) [8]: is the basis of many language learning models. It is widely used in many tasks that require text representation. Of course, this includes social event detection;
- **GraphMSE** (2021) [17]: is the start-of-the-art heterogeneous graph representation learning model with automatically selects meta-path;
- **FinEvent** (2022) [24]: is the start-of-the-art HIN social event detection model.

*5.1.3 Parameter and Model Settings.* We use 256-dimensional embeddings for the text model and 120-dimensional embeddings for the HIN model. For GraphHAM, we set the learning rate at 0.01, the neighborhood exploration limit number $\tau$ at 6 on the Kawarith and CrisisLexT6, 2 on the Twitter2012, and the meta-path length $\gamma$ at 3 on the Kawarith and CrisisLexT6, 2 on the Twitter2012. For TD-IDF [2], Word2Vec[3], FastText[4], Bert [5], GraphMSE [6], and FinEvent [7] we use the open-source implementations. In particular, for the representation learning models TD-IDF, Word2Vec, and Bert, we only use pre-trained models to extract the representation of text in the three datasets and directly divide the extracted representation into the same training ratio as we set before. Finally, enter the logistic regression classifier to classify the nodes without adding any

---

[2]https://scikit-learn.org/stable/modules/feature_extraction.html
[3]https://radimrehurek.com/gensim/models/word2vec.html
[4]https://fasttext.cc/docs/en/python-module.html
[5]https://huggingface.co/docs/transformers/model_doc/bert
[6]https://github.com/pkuliyi2015/GraphMSE
[7]https://github.com/RingBDStack/FinEvent

**Table 3: Comparison experiment results of node classification of all models.**

| | Kawarith | | | | | | | | | |
|---|---|---|---|---|---|---|---|---|---|---|
| Tr. Ratio | 5% | | 10% | | 20% | | 40% | | 70% | |
| Metrics | Micro F1 | Macro F1 | Micro F1 | Macro F1 | Micro F1 | Macro F1 | Micro F1 | Macro F1 | Micro F1 | Macro F1 |
| TF-IDF | 0.6285±0.0086 | 0.5217±0.1152 | 0.7791±0.0646 | 0.7193±0.0859 | 0.8382±0.0376 | 0.8125±0.0047 | 0.9105±0.0175 | 0.9015±0.0208 | 0.9287±0.0097 | 0.9249±0.0110 |
| Word2Vec | 0.5447±0.0252 | 0.4315±0.0223 | 0.5816±0.0048 | 0.4523±0.0055 | 0.5900±0.0045 | 0.4728±0.0130 | 0.6248±0.0052 | 0.5151±0.1260 | 0.6564±0.0080 | 0.5623±0.0091 |
| FastText | 0.2501±0.0000 | 0.6670±0.0000 | 0.2519±0.0001 | 0.6930±0.0004 | 0.5612±0.0027 | 0.3467±0.0011 | 0.7201±0.0000 | 0.5684±0.0023 | 0.8512±0.0019 | 0.8206±0.0032 |
| BERT | 0.6522±0.0122 | 0.6164±0.0181 | 0.6849±0.0097 | 0.6448±0.0104 | 0.7247±0.0043 | 0.7048±0.0067 | 0.7565±0.0072 | 0.7313±0.0075 | 0.7695±0.0092 | 0.7502±0.0101 |
| GraphMSE | 0.1720±0.0778 | 0.1232±0.0490 | 0.1382±0.0002 | 0.1216±0.0010 | 0.9105±0.0002 | 0.8966±0.0011 | 0.9330±0.0016 | 0.9218±0.0022 | 0.9470±0.0008 | 0.9400±0.0003 |
| FinEvent | 0.7986±0.0185 | 0.8001±0.0100 | 0.8632±0.0000 | 0.8566±0.0002 | 0.8898±0.0200 | 0.8813±0.0200 | 0.9066±0.0250 | 0.8964±0.0237 | 0.9259±0.0087 | 0.9136±0.0132 |
| GraphHAM | **0.8512±0.0036** | **0.8262±0.0047** | **0.8818±0.0134** | **0.8632±0.0152** | **0.9151±0.0052** | **0.9026±0.0073** | **0.9340±0.0029** | **0.9224±0.0040** | **0.9510±0.0041** | **0.9457±0.0053** |
| | CrisisLexT6 | | | | | | | | | |
| TF-IDF | **0.9064±0.0126** | **0.9063±0.0126** | **0.9216±0.0056** | **0.9209±0.0057** | **0.9265±0.0036** | **0.9255±0.0037** | **0.9316±0.0025** | **0.9307±0.0026** | 0.9365±0.0015 | 0.9356±0.0016 |
| Word2Vec | 0.7081±0.0062 | 0.7093±0.0068 | 0.7329±0.0057 | 0.7342±0.0047 | 0.7573±0.0027 | 0.7590±0.0030 | 0.7766±0.0053 | 0.7775±0.0053 | 0.7948±0.0043 | 0.7925±0.0034 |
| FastText | 0.6512±0.0072 | 0.6402±0.0120 | 0.8967±0.0000 | 0.8957±0.0000 | 0.9162±0.0002 | 0.9152±0.0001 | 0.9282±0.0004 | 0.9271±0.0004 | **0.9387±0.0009** | **0.9380±0.0009** |
| BERT | 0.7794±0.0054 | 0.7765±0.0058 | 0.7984±0.0053 | 0.7948±0.0055 | 0.8327±0.0018 | 0.8297±0.0017 | 0.8459±0.0028 | 0.8436±0.0029 | 0.8495±0.0038 | 0.8468±0.0036 |
| GraphMSE | 0.1475±0.0380 | 0.0997±0.0319 | 0.1538±0.0430 | 0.1136±0.0184 | 0.8911±0.0044 | 0.8656±0.0552 | 0.9065±0.0013 | 0.9061±0.0008 | 0.9100±0.0032 | 0.9094±0.0031 |
| GraphHAM | 0.8577±0.0220 | 0.8574±0.0219 | 0.8826±0.0094 | 0.8815±0.0096 | 0.9031±0.0010 | 0.9024±0.0017 | 0.9165±0.0017 | 0.9158±0.0017 | 0.9218±0.0027 | 0.9213±0.0023 |
| | Twitter2012 | | | | | | | | | |
| TF-IDF | 0.3059±0.0077 | 0.0721±0.0038 | 0.4384±0.0069 | 0.1527±0.0055 | 0.5610±0.0008 | 0.2295±0.0050 | 0.6393±0.0006 | 0.2935±0.0007 | 0.6789±0.0026 | 0.3405±0.0035 |
| Word2Vec | 0.4480±0.0063 | 0.1653±0.0053 | 0.5025±0.0019 | 0.2004±0.0023 | 0.5348±0.0040 | 0.2372±0.0050 | 0.5685±0.0039 | 0.2741±0.0051 | 0.5714±0.0054 | 0.2813±0.0047 |
| FastText | 0.0999±0.0000 | 0.0073±0.0000 | 0.0989±0.0004 | 0.0013±0.0000 | 0.1333±0.0008 | 0.0036±0.0000 | 0.1537±0.0001 | 0.0055±0.0000 | 0.1725±0.0000 | 0.0070±0.0000 |
| BERT | 0.4817±0.0031 | 0.2451±0.0056 | 0.5695±0.0056 | 0.3433±0.0067 | 0.6262±0.0041 | 0.4272±0.0055 | 0.6638±0.0062 | 0.4841±0.0176 | 0.6889±0.0032 | 0.5158±0.0052 |
| GraphMSE | 0.0015±0.0004 | 0.0006±0.0004 | 0.0017±0.0003 | 0.0009±0.0002 | 0.7494±0.0019 | 0.5633±0.0077 | 0.7856±0.0040 | 0.6343±0.0070 | 0.8057±0.0046 | 0.6716±0.0094 |
| GraphHAM | **0.6887±0.0154** | **0.4492±0.0129** | **0.7538±0.0105** | **0.5457±0.0006** | **0.8028±0.0022** | **0.6392±0.0016** | **0.8290±0.0016** | **0.6776±0.0042** | **0.8414±0.0013** | **0.7100±0.0040** |

additional operations. Furthermore, since FinEvent clusters events, our test metrics are for classification tasks. Therefore, we applied the method from [34] to map FinEvent clustering results to virtual labels and then converted them into a multi-classification problem. Note that we only tested the FinEvent model on the Kawarith dataset as out-of-memory issues occurred on the other two datasets. All experiments are conducted on a Macbook Air with an M1 chip. The result reports the mean and standard deviations of 5 times experiments.

## 5.2 GraphHAM Mode Performance Comparison (RQ 1 and RQ 2)

The experimental results of GraphHAM and baseline models are shown in Table 3. We set the training rate at 5%, 10%, 20%, 40%, and 70% for node classification task. For each training ratio, we set the validation ratio to 10% and the rest to the testing ratio.

In general, we can see that GraphHAM performs better than all baseline models on the Kawarith and Twitter2012 datasets. Only on the CrisisLexT6 dataset, its performance is worse than the TF-IDF and FastText, but it still achieves the best performance among complex models. The reason is that the TF-IDF and FastText models are more suitable for data sets with simple data structures and fewer event types. Especially on the Twitter2012 dataset, FastText is basically unable to learn any information. In addition, compared to the strong baseline model GraphMSE, our model improves on all percentages of the training set, especially at training ratios below 20%. At a training ratio of greater than or equal to 20%, it can also be 0.32%, 1.13%, and 4.42% higher than GraphMSE on the Kawarith, CrisislexT6 and Twitter2012 datasets respectively.

## 5.3 Ablation Studies

We conduct ablation experiments to verify the role of each component in the model, and we use a training ratio of 20% for experiments and report the mean and standard deviations of 5 times experiment results on the validation set.

*5.3.1 GraphHAM Performance in Different Space (RQ3).* We run GraphHAM-PoincareBall, GraphHAM-Hyperboloid, and GraphHAM-Euclidean and test their node classification performance. GraphHAM-PoincareBall means the model embeds nodes into hyperbolic space via the PoincareBall model. Similarly, the GraphHAM-Hyperboloid means model embeds nodes into the hyperbolic space through the Hyperboloid model. GraphHAM-Euclidean means the model embeds nodes into the Euclidean space. The performance is shown in Table 5. It is obvious that the hyperbolic space performs better than the Euclidean space, and, in general, the PoincareBall model in the hyperbolic space is more suitable for GraphHAM. Furthermore, we found that the more complex the data, the more hyperbolic space improves the model. For example, on the CrisisLexT6 and Kawarith datasets, hyperbolic space improves by 3% and 4% compared to Euclidean space. For the more complex Twitter2012 dataset, the hyperbolic space improved the model's performance by 7%. Therefore, we believe that the more complex the data, the deeper the tree-like structure of the data will be, and the more hyperbolic space can reduce distortion caused by data embedding.

*5.3.2 The impact of the text model and the HIN model (RQ4).* As mentioned before, GraphHAM has a text model as an auxiliary for heterogeneous models. We separately trained the text model and HIN model in the framework with the overall framework to compare the performance of node classification when the training ratio was 20%. The results are shown in Table 4. We can find that

**Table 4: The impact of different models in GraphHAM framework at** 10%, 20% **and** 40% **training ratio.**

| Training Ratio | 10% | | 20% | | 40% | |
|---|---|---|---|---|---|---|
| Kawarith | Micro-F1 | Macro-F1 | Micro-F1 | Macro-F1 | Micro-F1 | Macro-F1 |
| GraphHAM | **0.9002±0.0015** | **0.8765±0.0028** | **0.9248±0.0030** | **0.9062±0.0070** | **0.9313±0.0000** | **0.9236±0.0003** |
| GraphHAM w/o Text Model | 0.2478±0.0531 | 0.1704±0.0696 | 0.9109±0.0015 | 0.8913±0.0094 | 0.9217±0.0015 | 0.9045±0.0010 |
| GraphHAM w/o HIN Model | 0.8487±0.0015 | 0.8088±0.0016 | 0.8723±0.0075 | 0.8430±0.008 | 0.8798±0.0060 | 0.8592±0.0067 |
| CrisisLexT6 | Micro-F1 | Macro-F1 | Micro-F1 | Macro-F1 | Micro-F1 | Macro-F1 |
| GraphHAM | **0.9099±0.0049** | **0.8861±0.0010** | **0.9116±0.0016** | **0.9021±0.0015** | **0.924±0.0021** | **0.9206±0.0013** |
| GraphHAM w/o Text Model | 0.1545±0.1108 | 0.1019±0.0792 | 0.9016±0.0024 | 0.8353±0.0870 | 0.9119±0.0020 | 0.9113±0.0005 |
| GraphHAM w/o HIN Model | 0.8332±0.0008 | 0.8293±0.0008 | 0.8692±0.0037 | 0.8665±0.0039 | 0.8841±0.0066 | 0.8844±0.0068 |
| Twitter2012 | Micro-F1 | Macro-F1 | Micro-F1 | Macro-F1 | Micro-F1 | Macro-F1 |
| GraphHAM | **0.7478±0.0045** | **0.5918±0.0540** | **0.8081±0.0011** | **0.6406±0.0016** | **0.8354±0.0015** | **0.6786±0.0021** |
| GraphHAM w/o Text Model | 0.0031±0.0015 | 0.0003±0.0002 | 0.7663±0.0041 | 0.6028±0.0013 | 0.8133±0.0026 | 0.7091±0.0074 |
| GraphHAM w/o HIN Model | 0.4677±0.0032 | 0.2147±0.0015 | 0.5003±0.0002 | 0.2131±0.0020 | 0.4988±0.0059 | 0.2159±0.0028 |

**Table 5: GraphHAM performance in different space.**

| Kawarith | Micro-F1 | Macro-F1 |
|---|---|---|
| GraphHAM-PincareBall | **0.9174±0.0015** | **0.9035±0.0008** |
| GraphHAM-Hyperboloid | 0.9142±0.0182 | 0.8977±0.0136 |
| GraphHAM-Euclidean | 0.8734±0.0121 | 0.8628±0.0264 |
| CrisisLexT6 | Micro-F1 | Macro-F1 |
| GraphHAM-PincareBall | 0.9035±0.0137 | 0.8924±0.0135 |
| GraphHAM-Hyperboloid | **0.9193±0.0008** | **0.9069±0.0004** |
| GraphHAM-Euclidean | 0.8882±0.0165 | 0.8735±0.0209 |
| Twitter2012 | Micro-F1 | Macro-F1 |
| GraphHAM-PincareBall | **0.8067±0.0034** | **0.6347±0.0040** |
| GraphHAM-Hyperboloid | 0.7602±0.0004 | 0.6052±0.0016 |
| GraphHAM-Euclidean | 0.7333±0.0118 | 0.5787±0.0107 |

**Table 6: Nodes embedding combination methods.**

| Kawarith | Micro-F1 | Macro-F1 |
|---|---|---|
| GraphHAM-SUM | 0.9013±0.0030 | 0.8706±0.0011 |
| GraphHAM-MEAN | **0.9163±0.0060** | **0.9069±0.0020** |
| GraphHAM-CONCAT | 0.8994±0.0242 | 0.8756±0.0353 |
| CrisisLexT6 | Micro-F1 | Macro-F1 |
| GraphHAM-SUM | 0.9066±0.0062 | 0.8991±0.0050 |
| GraphHAM-MEAN | 0.9122±0.0016 | **0.9021±0.0008** |
| GraphHAM-CONCAT | **0.9131±0.0012** | 0.8388±0.0739 |
| Twitter2012 | Micro-F1 | Macro-F1 |
| GraphHAM-SUM | **0.8325±0.0064** | **0.6784±0.0001** |
| GraphHAM-MEAN | 0.8086±0.0016 | 0.6388±0.0007 |
| GraphHAM-CONCAT | 0.8274±0.0026 | 0.6803±0.0015 |

when we train one of the models alone, the performance is not as good as when they are trained together. The HIN model accounts for a larger proportion than the text model at 20% training ratio. However, the role of these two models cannot be clearly seen by looking at the 20% training ratio, so we have added the ratios near the 20% ratio: 10% and 40%. We can find that 20% exists as an interval point. When the training ratio is less than 20%, text models play a key role, and the HIN model is basically unable to learn features. When the training ratio reaches or even exceeds 20%, the HIN model begins to play a role and takes a dominant position, and the text model plays an auxiliary role in helping the entire framework learn information that the HIN model cannot capture.

*5.3.3 The impact of the embedded combination of "SUM", "MEAN", and "CONCAT" on the model (R5).* We run GraphHAM-SUM, GraphHAM-MEAN, and GraphHAM-CONCAT to see the performance of node classifications on the three datasets. Table 6 shows the result of the experiments. It can be seen that for the Kawarith and CrisisLexT6 datasets, the "MEAN" method performs better, but for the Twitter2012 dataset, the "SUM" is the best-performing node representation combined method.

For such results, we believe that the merged nodes embedding method's performance is related to the data's complexity. For data of simple complexity, such as the Kawarith and CrisisLexT6 datasets, the "CONCAT" or "MEAN" method is more suitable for our framework. Let's review Table 4 again. At a training rate of 20%, the

text and HIN models extract sufficient features. This is why the "SUM" or "MEAN" method can perform well in these two datasets: they can better extract the common features in the two models. However, for the complex dataset Twitter2012, we can see from the table that the performance of the text model and HIN model is not as good as the above two datasets. This also means that they extracted a few common features. If we use "MEAN" or "MEAN" to combine this set of features, some features will be lost for the entire framework. Therefore, the "SUM" method can better extract features for complex data.

## 5.4 Hyperparameter Analysis (RQ6)

We analyze the hyperparameters of the model, which are the number of neighbor explorations around a node $\tau$ and the length of the meta-path $\gamma$. We set $\tau = 1, 2, 3, 4, \cdots, 9, 10$, the result shown in Fig.2(a). We found that, in general, the model with $\tau$ around 5 performs satisfactorily in all the datasets. Specifically, for the Kawarith and CrisisLexT6 datasets, when $\tau \leq 6$, the model's performance shows an increasing trend, and when $\tau > 6$, the model's performance fluctuates. For the Twitter2012 dataset, the best performance is when $\tau = 2$, but the performance when $\tau \leq 5$ is also satisfactory, and when $\tau > 5$, there is a downward trend.

In addition, we analyze the length of the meta-path $\gamma$. The time complexity of the model will increase exponentially as $\gamma$ increases. What we want is to explore how long $\gamma$ is required for our model to

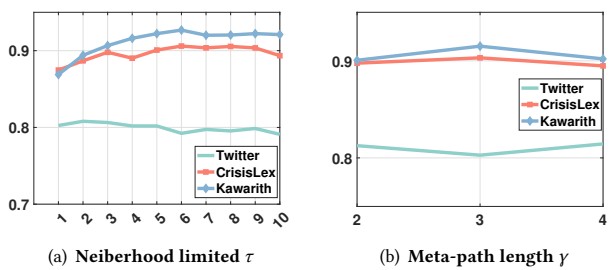

(a) Neiberhood limited $\tau$      (b) Meta-path length $\gamma$

**Figure 2: Performances with different parameters in all datasets at 20% training rate.**

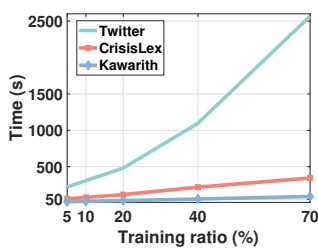

**Figure 3: The relationship between training ratio and training time.**

be satisfactory. The result is shown in the Fig.2(b). We found that on the Kawarith and CrisislexT6 datasets, $\gamma = 3$ allowed the model to achieve satisfactory performance. On the Twitter2012 dataset, $\gamma = 2$ can make the model perform well.

### 5.5 Time Efficiency(RQ7)

**Table 7: Time efficiency analysis for all models at 20% training ratio.**

| | TF-IDF | Word2Vec | FastText | Bert | GraphMSE | FinEvent | GraphHAM |
|---|---|---|---|---|---|---|---|
| *Kawarith* (4,860) | | | | | | | |
| Time | 0.93s | 0.95s | 0.75s | 480.17s | 7.53s | 310.21s | 34.22s |
| *CrisisLexT6* (18,157) | | | | | | | |
| Time | 1.15s | 2.82s | 7.18s | 960.42s | 15.12s | –– | 116.75s |
| *Twitter2012* (68,841) | | | | | | | |
| Time | 3.71s | 21.84s | 125.52s | 1581.62s | 34.57s | –– | 480.41s |

We mainly analyze the efficiency of our model from two aspects. The first is the time required for our model training to reach convergence for different training ratios. As shown in Fig. 3, we can see that in the smaller datasets, the Kawarith (4,860 tweets) and the CrisisLexT6 (18,157 tweets), our model's training time increases slowly as the training ratio increases. However, in the Twitter2012 (68,841 tweets) dataset, when the training ratio exceeds 20%, the training time will show a trend close to exponential growth.

Then, we run all models on all datasets with a training ratio of 20%, and the training time is until the models reached convergence. The results are shown in Table 7. We can see that their training time is very fast for simple models such as TF-IDF, Word2Vec, and FastText. However, compared with complex models such as Bert and FinEvent, our model still has great efficiency improvements. At 20% training ratio, the training time of our model is still slightly inferior to GraphMSE, but we have better performance, especially when the training ratio is less than 20%.

### 5.6 Dataset Complexity

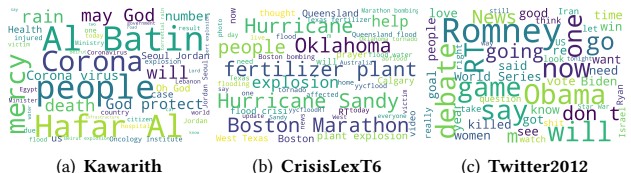

(a) **Kawarith**     (b) **CrisisLexT6**     (c) **Twitter2012**

**Figure 4: Datasets' word cloud.**

In this section, we discuss the complexity of the dataset we used, which has implications for many components of our model. We apply the word cloud with the same settings to show the concentration of keywords in each dataset as shown in Fig.4. The word cloud represents the frequency of word occurrence through the size of the font and displays the frequency of different words in the data through the number of words appearing per unit area. We can see that only a few high-frequency words appear in the Kawarith data, and the words in the entire data are also sparse. The high-frequency words in CrisisLexT6 data are average and not sparse. This is why TF-IDF and FastText models perform outstandingly on CrisisLexT6 data. In addition, the different words in the Twitter2012 dataset have different frequencies, which reflects that this dataset is more complicated than the above two datasets and results in the need to learn heterogeneous information to capture more features. This is the main reason why TF-IDF and FastText perform poorly on the Twitter2012 dataset.

## 6 CONCLUSION

In this article, we propose a social media detection model that combines the automatic selection of meta-path and hyperbolic space representation: GraphHAM. Specifically, we apply hyperbolic space representation to learn tree-structured data in the HIN environment and use efficient sampling technology to improve model efficiency greatly. Experimental results show that our model is highly competitive with existing social event detection models in terms of model performance and model efficiency. In addition, our model can perform well on all datasets we used and with a relatively low training ratio. Our model can achieve satisfactory performance when the training ratio reaches 20%. In the future, we plan to extend the model to online social media detection environments, including how to face incremental data streams and imbalanced data sets.

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
