# OpenReview forum: "An Efficient Automatic Meta-Path Selection for Social Event Detection via Hyperbolic Space"
_ACM.org/TheWebConf/2024/Conference — TheWebConf24_

### Official Review · Reviewer_coMn · 2023-11-20

**Novelty:** 4
**Technical Quality:** 4

**Review:**

The paper addresses the challenge of timely detecting social events, such as natural disasters, on social media to assist communities in avoiding harm. Existing methods often rely on fixed sets of meta-paths and expensive labeling, limiting their effectiveness. To overcome these challenges, the authors propose the GraphHAM model, which combines Hyperbolic space and automatic meta-path selection. This approach efficiently selects meta-paths, converts them into vectors, and employs a novel Hyperbolic Multi-Layer Perceptron to capture semantic and structural information in social messages. Extensive experiments demonstrate that GraphHAM achieves outstanding performance on real-world data, using just 20% of the dataset for training, making it a promising solution for social event detection.

Strengths:
+ A wide range of evaluations: 7 research questions, 3 datasets, 6 baselines, performance comparison, ablation studies, time efficiency, parameter analysis.
+ Clear illustration of the proposed framework
+ Fast training time despite of graph representation learning.

Weaknesses:
+ Motivation is not clear. It would be great to clarify a hyperbolic space is needed for capturing a large amount of unlabeled data. Some motivating examples might help.
+ The novelty is not clear. It is mentioned in the related work that the proposed framework applies hyperbolic space for social event detection in heterogeneous social networks. What would be the challenges to do so?
+ The intuition of the proposed framework is not clear. It is mentioned that the meta-paths are enumerated from the selected points instead of all nodes to reduce the complexity. But what are selected points?
+ The contribution seems to be incremental. It combines the meta-path sampling of GraphMSE, graph hyperbolic embedding, and some attention mechanisms to capture the tree-like structure of information diffusion on social networks.
+ Problem formulation is not clear. It would be great to clarify which social items we are detecting here (could not find a clear problem statement in Section 3).

Updated: I have read the author response.

**Questions:**

+ Motivation is not clear. It would be great to clarify a hyperbolic space is needed for capturing a large amount of unlabeled data. Some motivating examples might help.
+ The novelty is not clear. It is mentioned in the related work that the proposed framework applies hyperbolic space for social event detection in heterogeneous social networks. What would be the challenges to do so?
+ The intuition of the proposed framework is not clear. It is mentioned that the meta-paths are enumerated from the selected points instead of all nodes to reduce the complexity. But what are selected points?
+ The contribution seems to be incremental. It combines the meta-path sampling of GraphMSE, graph hyperbolic embedding, and some attention mechanisms to capture the tree-like structure of information diffusion on social networks.
+ Problem formulation is not clear. It would be great to clarify which social items we are detecting here (could not find a clear problem statement in Section 3).
+ Section 5.6 seems to belong in experimental settings as it describes some dataset statistics or we might need a separate discussion section.
+ It is not clear why the model still performs well with small amount of training data.
+ It would be great to summarize the end-to-end training and testing processes.

**Reviewer Confidence:**

3: The reviewer is confident but not certain that the evaluation is correct

**Scope:**

3: The work is somewhat relevant to the Web and to the track, and is of narrow interest to a sub-community

---

### Official Review · Reviewer_i1TH · 2023-11-20

**Novelty:** 4
**Technical Quality:** 5

**Review:**

The authors propose a methodology (GraphHAM) for supervised learning of message topics in on-line message systems. The claimed key novel algorithmica ideas are the use of automatic meta-path selection in heterogeneous communication networks, and the use of hyperbolic space embeddings.
The novel algorithm is then tested in three data sets with a minimum of 4.8K and a maximum of 68K nodes , with respectively 6,6 and 503 labeled topics. All datasets have nodes of 3 types (tweet, entity, and user). These are then mapped to graphs with a number of nodes tranging in 9.7K to 98K and number of edges from 14k to 148K.
The competitors are 2 baselines (tf-idf and word2vec9 as well as 4 more advanced recent methods.
Note that one of these methods, GraphMSE, also has automatic meta-path selection.
The experimental comparative performance results are shown in table 3. The authors note correctly that in one dataset (CrisisLexT6) the basic TF-IDF consistently outperforms GraphHAM.
On the two remaining datasets GraphHAM has marginal gains on GraphMSE for size of the training set above 20\%.
Interestingly GraphHAM has acceptable performance (markedly superior to all compared methods) for size of the training set below 20\%.
On the negative side:
1) Overall there is a lack of novelty, the adaptive meta-path selection is used also in GraphMSE (ref [21]) and the use of hyperbolic embeddings is known to be beneficial. See
Balazevic, Ivana, Carl Allen, and Timothy Hospedales. "Multi-relational poincaré graph embeddings." Advances in Neural Information Processing Systems 32 (2019).
Nickel, Maximillian, and Douwe Kiela. "Poincaré embeddings for learning hierarchical representations." Advances in neural information processing systems 30 (2017). The authors should compare their approach with that in those two papers form a structural/algorithmic point of view.

2) The results of table 3 are not sufficient to support the claims of superiority of the proposed method.
On the positive side:
1) A good performance with a small training set is rather unusual. it is noted by the authors, but probably more research and experiments are needed to explore the full potential of this interesting phenomenon, as reported in R2, R3 and R4.

**Questions:**

Table 2 shows that for the 3 test cases adopted there are 4 possible meta-paths possible. In this setting even a simple exhaustive search would work well. I wonder if the authors did consider other data sets in which the number of possible meta-paths is much larger and thus would benefit greatly from an automatic efficient selection strategy.

**Ethics Review Description:**

no issue

**Reviewer Confidence:**

3: The reviewer is confident but not certain that the evaluation is correct

**Scope:**

4: The work is relevant to the Web and to the track, and is of broad interest to the community

---

### Official Review · Reviewer_opCk · 2023-11-24

**Novelty:** 5
**Technical Quality:** 4

**Review:**

The author proposes an efficient heterogeneous information graph learning framework, with the model's primary advantage lying in achieving optimal performance with less data, as demonstrated in the experimental results. The claimed reasons for this advantage are Meta-path Sampling and Hyperbolic MLP. While the overall text is readable, I have concerns regarding the necessity of and performance enhancement strategies for the two introduced components. These concerns are highlighted in the following aspects:

**Model Section:**

1. The author claims to have implemented automatic meta path selection, but based on the introduction and context in section 4.2, I do not understand where "automatic" is reflected. Additionally, there is a lack of ablation experiments (w/o automatic meta path selection) to demonstrate the impact on performance.

2. Regarding Hyperbolic MLP, it seems the author employs a specific pattern to enhance model performance. However, a comparison with traditional MLP in the experimental context would be valuable to elucidate the differences.

**Experimental Section:**

3. The ablation experiments lack a comparison of key components. (see Q1 and Q2)

4. The current model might be an improvement over GraphMSE based on experimental results. I hope you can provide a direct comparison with GraphMSE.

5. The advantage of the model is that using fewer datasets can achieve better results. This advantage mainly lies in reducing the time required for dataset construction (such as data annotation time)? If that's the case, I need to understand the time comparison between building a 20% and 100% dataset. If the time is relatively close or the dataset is relatively easy to construct, I believe the author's advantage will be reduced.

6. Performance comparison using a 100% dataset is recommended for a comprehensive evaluation.

7. In Experiment 5.5, the author claims that more efficient models exhibit higher efficiency, but the author's model is slower than all other models except Bert. The inefficiency of larger models is attributed to a large number of parameters, which is advantageous with sufficient data. However, as the comparison is conducted with less data, the proposed model does not appear to be efficient when compared to models with fewer parameters.

**Questions:**

Refer to the above questions

**Reviewer Confidence:**

2: The reviewer is willing to defend the evaluation, but it is likely that the reviewer did not understand parts of the paper

**Scope:**

4: The work is relevant to the Web and to the track, and is of broad interest to the community

---

### Official Review · Reviewer_eftV · 2023-11-25

**Novelty:** 5
**Technical Quality:** 5

**Review:**

This paper focuses on event detection on social media platforms. The authors propose a graph-based approach that considers different aspects of online activity to result in an improved performance. In particular, a Heterogeneous Information Graph representation via Hyperbolic space combined with an Automatic Meta-path selection model is proposed, which succeeds to mitigate the limitations of previous similar solutions.

Overall, the authors present clearly from the beginning the motivation behind the proposed solution. In addition, the Related Work section clearly presents the existing deficiencies that need to be addressed in this work. Finally, various experiments have been conducted on three datasets in order to demonstrate the effectiveness of the proposed solution, as well as the individual components that make it up.

As two minor comments. First, I would like to see the discussion of "Dataset Complexity" earlier in the paper, as many observations made in the experimental results section are highly related to the complexity of datasets (the paper will be updated accordingly; see R1) . Additionally, it would be nice if the authors mentioned how they extracted the named entities from the input data (clarified; see R2).

Finally, I would appreciate it if the authors would make the code publicly available upon acceptance, as this would make it easier for others to reproduce the results (the code will be made publicly available after acceptance; see R3).


&nbsp;

Update: I acknowledge that I have read the authors' responses.

**Questions:**

- Do the authors intend to make their code publicly available upon acceptance?

**Reviewer Confidence:**

3: The reviewer is confident but not certain that the evaluation is correct

**Scope:**

4: The work is relevant to the Web and to the track, and is of broad interest to the community

---

### Official Review · Reviewer_KJr1 · 2023-11-26

**Novelty:** 3
**Technical Quality:** 4

**Review:**

This paper studies the problem of social event detection, and propose an efficient heterogeneous information graph learning framework GraphHAM. The proposed framework can automatically select meta-path and combine hyperbolic space to learn information on social media. The experimental results on several datasets demonstrate that GraphHAM can achieve outstanding performance.

S1: The authors present a heterogeneous information graph learning framework GraphHAM, which can select meta-path automatically and learn hyperbolic spaces representation. GraphHAM can effectively handle social media detection, even the training data is limited.
S2: The paper is well organized and easy to follow.
S3: The problem addressed in this paper is important on social media. The automatic meta-path selection technology and hyperbolic space representation can provide a meaningful reference for interested researchers.

W1: The paper utilizes many existing technologies, and the contribution of originality is relatively small.
W2: Compared to some existing methods, the experimental results of the paper do not demonstrate obvious advantages.

**Questions:**

Q1: This paper draws upon various established methods while presenting a limited number of original contributions.
Q2: Construct heterogeneous information networks in data preprocessing seems to require manual intervention, which appears to be a relatively complex process.
Q3: The experiment results did not demonstrate superior outcomes across all evaluated datasets, hence posing challenges in demonstrating its effectiveness.

**Reviewer Confidence:**

3: The reviewer is confident but not certain that the evaluation is correct

**Scope:**

4: The work is relevant to the Web and to the track, and is of broad interest to the community

---

### Decision · Program_Chairs · 2024-01-22

**Decision:**

Accept

**Comment:**

The paper proposes a graph learning framework for the detection of events.
 The writing is clear, well-motivated and experiments are convincing.
 Some concerns were raised about lack of novelty, marginal gain and some missing baselines.
 Most of the reviewers' comments were addressed during the rebuttal phase, and the authors did a very
 convincing job in explaining the critical aspects of their work. I suggest that they include
 part of the discussion in the final version.